# Context-dependent activation and evolutionary buffering of a mating pheromone in fission yeast
Taisuke Seike [1,2] ✉, Natsue Sakata[2], Hazuki Kotani[2] & Chikara Furusawa [2,3]

The evolution of mating signals drives reproductive isolation and speciation across diverse lineages. However, how short peptide pheromones, typically subject to strong structural constraints, achieve functional diversification remain unclear. In the fission yeast *Schizosaccharomyces pombe*, a previously established library of 153 single-amino acid variants of the mating pheromone M-factor was applied to large-scale competition assays under varied mating conditions. Mutations deleterious under standard conditions became advantageous at specific environmental pH levels, demonstrating context-dependent pheromone function activation. Synthetic peptide assays confirmed that certain substitutions act as environmental molecular switches. Comparative analysis with the closely related *Schizosaccharomyces octosporus* species further identified a permissive mutation that mitigates the effects of otherwise inactivating changes, enabling an evolutionary route without intermediate fitness loss. Our findings reveal how short peptide signals can evolve via environmentally contingent activation and compensatory interactions, offering a mechanistic framework for understanding the ecological and evolutionary dynamics of mating communication.

The early evolution of reproductive isolation is a central yet experimentally underexplored topic in evolutionary biology. Although postzygotic barriers are often studied retrospectively in diverged lineages[1,2], the initial stages of prezygotic isolation, particularly how mate recognition systems respond to environmental heterogeneity, remain poorly understood. Environmental variation in factors such as pH, temperature, and resource availability can modulate the efficacy of mating signals, promoting assortative mating and divergence[3]. Such effects are documented in insects, amphibians, and fishes, where habitat-specific sensory conditions alter mate choice[4]. However, the molecular mechanisms enabling mating systems to evolve under such conditions remain elusive.

Fungal pheromone–receptor systems offer a tractable model, with clear parallels to animal chemical communication[5,6]. Variations in pheromone structure or expression can promote assortative mating and prezygotic reproductive isolation[7–9]. As many sexual pheromones bind their receptors with high specificity, strong structural constraints exist, and single amino acid changes in short peptides can abolish function[10–13]. This raises key questions about how much variation these molecules can tolerate, and the ecological conditions that favor it.

Yeasts have long served as models for studying the molecular evolution of mating systems[14]. Comparative studies in budding yeasts reveal that pheromone–receptor diversity reflects a balance between strict recognition and evolutionary flexibility[10,15]. Experimental evolution shows that pheromone receptor mutations can rapidly enhance species discrimination, preventing maladaptive hybridization[16,17]. In the fission yeast *Schizosaccharomyces pombe*, single amino acid changes in the M-factor peptide or its receptor Map3 can cause complete mating incompatibility, indicating that coevolution can establish new reproductive barriers[12]. However, previous work has not examined such variants under ecological competition or variable environments. A plausible scenario is that environmental variability reshapes the signaling landscape, favoring different variants in different microhabitats, whereas additional permissive mutations allow otherwise deleterious changes to be retained, collectively driving divergence.

Herein, we used *S. pombe*, which alternates between asexual growth and nitrogen depletion-induced sexual reproduction[18]. M-type cells secrete a nine-amino-acid farnesylated pheromone (M-factor), and P-type cells secrete a distinct peptide (P-factor); each activates its specific receptor, Map3 or Mam2[19]. Natural mating success likely depends on both compatibility and environmental modulation of activity.

A comprehensive M-factor variant library was generated by introducing all 19 possible substitutions at each of eight non-cysteine residues of the nine-amino-acid mating pheromone M-factor (YTPKVPYMC$^{Far}$-OCH$_3$)[11]. In total, 152 mutants plus wild type were pooled and subjected to repeated mating–germination cycles under varied media and pH. Amplicon

[1]Department of Bioscience and Bioinformatics, Kyushu Institute of Technology, Fukuoka, Japan. [2]Center for Biosystems Dynamics Research, RIKEN, Kobe, Hyogo, Japan. [3]Universal Biology Institute, The University of Tokyo, Tokyo, Japan. ✉e-mail: seike@bio.kyutech.ac.jp

sequencing quantified variant abundance, revealing several with context-dependent advantages, particularly at low or high pH. Synthetic peptide assays confirmed that the P6H variant activated Map3 preferentially at higher pH values, consistent with an environmentally responsive switch. In *S. octosporus*, which differs from *S. pombe* by three M-factor residues (T2Q, V5P, Y7A)[20], the T2Q substitution, identified in the variant library as a context-dependent allele, buffered deleterious changes as a permissive mutation. This suggests an evolutionary path for retaining function without non-functional intermediates.

Our findings provide a mechanistic framework showing how short, structurally constrained peptides evolve via environmental modulation and mutational buffering. By integrating mutational scanning, ecological selection, and receptor activation profiling, principles driving diversification of molecular signaling systems under natural selection were identified.

## Results

### Establishing a high-throughput competition assay using comprehensive M-factor variants

The M-type pheromone (M-factor) in *Schizosaccharomyces pombe* is critical for sexual reproduction, yet how sequence variation influences mating-related fitness remains unclear. To address this, a previously constructed library of 152 single-amino acid variants was used[11], each carrying a substitution at one of the eight non-cysteine residues in the mature peptide (YTPKVPYMC$^{Far}$-OCH$_3$). In total, 153 $h^{90}$ strains, including the wild type, were combined at equal abundance (~$1.0 \times 10^5$ cells per strain; ~$1.5 \times 10^7$ cells in total) at the start of the experiment (Fig. 1a).

All strains shared the $h^{90}$ genetic background, enabling autonomous mating-type switching between M-type (pheromone-producing) and P-type (pheromone-receiving) states. This ensured that each variant experienced an identical receptor environment during the P-type phase, such that relative fitness differences reflected primarily M-factor function during the M-type phase. Each mating–sporulation cycle was initiated by inoculating the mixed population onto solid media (Malt Extract Agar (MEA), Edinburgh Minimal Medium 2 (EMM2), Pombe Minimal Glutamate (PMG), or Synthetic Sporulation Agar (SSA)) under varied pH (4.0–7.0) and temperature (30–35 °C). After 4 days (sufficient for sporulation) vegetative cells were eliminated with 30% ethanol[21], and the resulting spores germinated in Yeast Extract Liquid (YEL) medium for 2 days before the next cycle (Fig. 1a).

To capture selection dynamics on different timescales, genomic DNA was sampled after cycle 1 and 5. The cycle 1 sample reflected the outcome of a single sexual reproduction round, detecting short-term selection on variant frequencies, whereas the cycle 5 sample represented cumulative effects across repeated cycles, revealing longer-term competitive consequences. Variant abundances were quantified using amplicon sequencing of a 24-base pair region within *mfm1*, which encodes the mature M-factor peptide (see Methods).

On MEA, substitutions at C-terminal residues (V5, P6, Y7, M8) were strongly depleted after both cycles (Fig. 1b and Supplementary Fig. 1a), consistent with reduced mating efficiency[11]. In contrast, most T2 substitutions increased in frequency (Fig. 1b), in some cases surpassing wild type, indicating the native sequence is not universally optimal. Similar trends occurred on EMM2 and PMG (Supplementary Fig. 2a, b), likely reflecting their comparable nutrient compositions. In contrast, SSA showed weaker selection, with many variants maintaining intermediate frequencies (Supplementary Fig. 2c), suggesting that medium components buffer the functional impact of M-factor variation via effects on nutrient diffusion, cell–cell contact, or pheromone degradation.

On non-sporulating YEA medium, control experiments yielded largely stable variant frequencies over five cycles (Fig. 1c and Supplementary Fig. 1b), with only modest changes observed for a small number of variants. These minor fluctuations may reflect subtle differences in vegetative growth or stochastic variation inherent to pooled assays. Notably, V5M variant was a clear exception, consistently declined under all conditions (Fig. 1b, c and Supplementary Figs. 1–6). This pattern is consistent with a general fitness

disadvantage, although technical contributions such as amplifications or sequencing bias cannot be completely excluded. All experiments were performed in triplicate, with high reproducibility (Supplementary Fig. 7). To confirm that single-strain mating efficiencies measured in isolation predict pooled competitive performance, post-competition frequencies after five cycles with theoretical expectations based on individual mating efficiencies were compared. The two were in close agreement (Supplementary Fig. 8), consistent with *S. pombe*'s inability to mate using exogenous pheromones alone[22]. This supported that each strain's competitive outcome is primarily determined by its own mating capacity rather than pheromone signals from competitors. Collectively, these results showed that our high-throughput competition assay robustly captures environmentally modulated pheromone variant selection within multiclonal populations, providing a powerful framework for quantifying mating-related fitness under diverse ecological conditions.

### Environmental pH modulates M-factor variant fitness and receptor activation

Peptide pheromone activity can be sensitive to environmental factors such as pH and temperature, which influence mating efficiency by altering molecular interactions or signal transduction. To investigate how these abiotic conditions modulate M-factor variant fitness, selection outcomes were systematically compared under temperature and pH regimes.

The impact of temperature was first assessed by conducting competition assays on EMM2 medium at 30 °C and 35 °C. As the restrictive temperature of *S. pombe* is 35.5 °C[23], 35 °C represents a physiologically relevant upper limit. No significant differences in variant frequencies were observed between the two conditions (Supplementary Fig. 3a–c).

Next, how environmental pH influences selection among M-factor variants was examined. EMM2 and SSA media were adjusted to pH 4.0, 5.5, and 7.0. Variant frequencies after five mating–sporulation cycles were compared with those at pH 5.5, serving as the baseline reference condition. On EMM2, no variants gained a selective advantage at pH 4.0 or 7.0, indicating this medium does not enhance pH-dependent pheromone efficacy, with several variants showing decreased performance at non-neutral pH (Supplementary Figs. 4, 5). In contrast, SSA exhibited marked pH effects; for example, the P6D variant was enriched at pH 4.0, whereas P6H and Y7W markedly increased at pH 7.0 (Fig. 2a, b and Supplementary Fig. 6). Notably, P6H showed a ~30-fold increase in relative frequency at pH 7.0 compared with pH 5.5, indicating a strong advantage at higher pH values (pH 7.0 under the screening conditions).

These results demonstrated that specific amino acid substitutions in the M-factor peptide confer conditional fitness benefits under distinct pH environments. This context-dependent selection reveals how environmental heterogeneity, such as local pH variation, can shape evolutionary trajectories of signaling alleles. Furthermore, variants such as P6H may function as "cryptic alleles": phenotypically neutral or even deleterious under standard conditions, yet highly advantageous in alternative environments. Such cryptic functional potential may provide a reservoir of latent adaptability, positioning populations for rapid evolutionary responses to environmental change.

### pH-dependent mating efficiency and receptor activation in individual M-factor variants

To validate whether pH-dependent shifts in strain frequencies observed in the competition assay reflect differences in mating success, mating efficiency was directly quantified for three environmentally sensitive M-factor variants —P6D, P6H, and Y7W—under defined pH conditions. Each strain was cultured on SSA plates adjusted to pH values ranging from 4.0 to 9.0 in 0.5-unit increments. Mating efficiency was evaluated after 48 h by microscopically scoring over 1000 cells per condition. The results broadly reflected the competitive trends; however, they revealed striking variant-specific pH profiles. P6D showed the highest mating efficiency at acidic pH, with frequencies at pH 4.0–5.0 exceeding those at pH 5.5 by more than two-fold; however, it failed to produce asci at pH ≥6.5 (Fig. 3a). In contrast, P6H

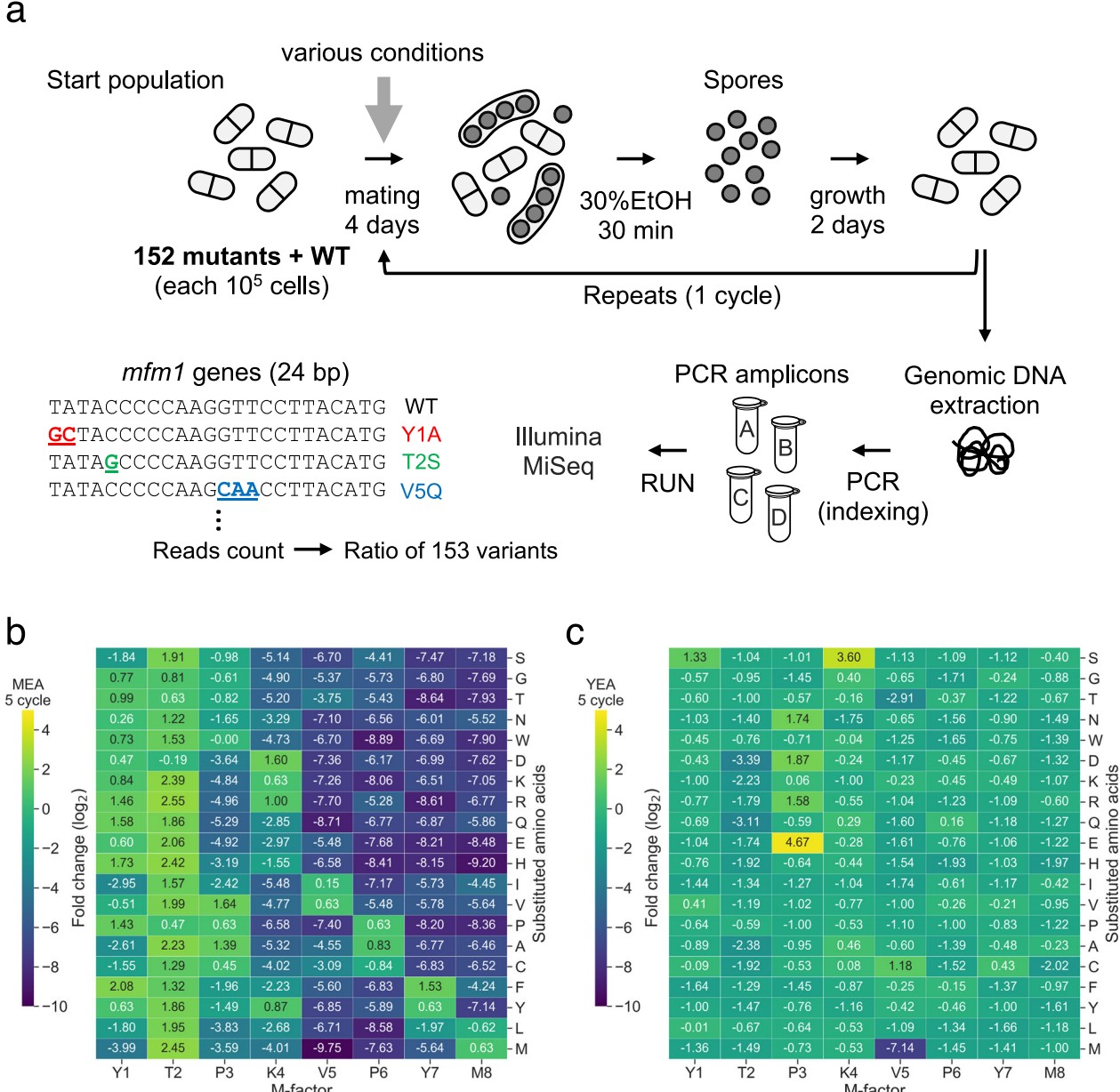

**Fig. 1 | High-throughput competition assay for assessing fitness effects of M-factor variants under mating and non-mating conditions. a** Schematic of the experimental workflow. In total, 153 $h^{90}$ strains, each carrying a unique single amino acid substitution in the mature M-factor peptide (YTPKVPYMC$^{Far}$-OCH$_3$), were mixed in equal proportions ($10^5$ cells per strain) and spotted on agar-based mating media under various environmental conditions. After 4 days of mating and sporulation, vegetative cells were selectively eliminated using 30% ethanol treatment for 30 min. Spores were germinated in liquid YEL medium for 2 days, completing one mating–germination cycle. Genomic DNA was extracted after cycles 1 and 5, and a 24-base pair region of *mfm1* encoding the mature M-factor was PCR-amplified.

Amplicon sequencing (75 bp reads, MiSeq) quantified relative variant abundance from sequence-specific read counts. **b** Heatmap showing $\log_2$ fold changes in strain abundance between cycles 1 and 5 on MEA (pH 5.5) averaged across biological replicates ($n = 3$). Variants with substitutions at C-terminal residues (V5, P6, Y7, M8) were strongly depleted, consistent with impaired mating activity. **c** Control assay on non-mating YEA medium, where strains were serially passaged without ethanol treatment. Most variants maintained stable frequencies indicating no major defect in vegetative growth. A consistent decrease in P3E and V5M abundance suggests a technical bias (e.g., PCR or sequencing). Values are mean of three replicates ($n = 3$).

reached maximal performance at pH 7.5–8.5, with approximately 50% of cells forming mature asci, despite being virtually sterile at pH 5.5, indicating a marked shift in mating competence across a narrow pH range (Fig. 3a, b). Y7W maintained consistently high sporulation across the entire pH gradient, similar to wild type (Fig. 3). Although Y7W enrichment at pH 7.0 in competition assays remains mechanistically unclear, its sustained mating capacity irrespective of pH may underlie this pattern. Collectively, these findings provided direct evidence that mating capacity in P6H is strongly modulated by environmental pH, with a marked shift from sterility to high performance across a narrow pH window.

Although the high-throughput competition assay quantifies relative lineage success after mating and sporulation, it does not reveal which specific steps drive fitness differences. Therefore, microscopy-based assays were used to directly measure zygote and ascus formation under defined pH conditions, separating potential effects on signaling, partner choice, and sporulation. These analyses were designed to determine whether environmentally sensitive variants identified in competition assays, particularly those at the P6 position, exhibited context-dependent effects on receptor activation, mating behavior, and sporulation.

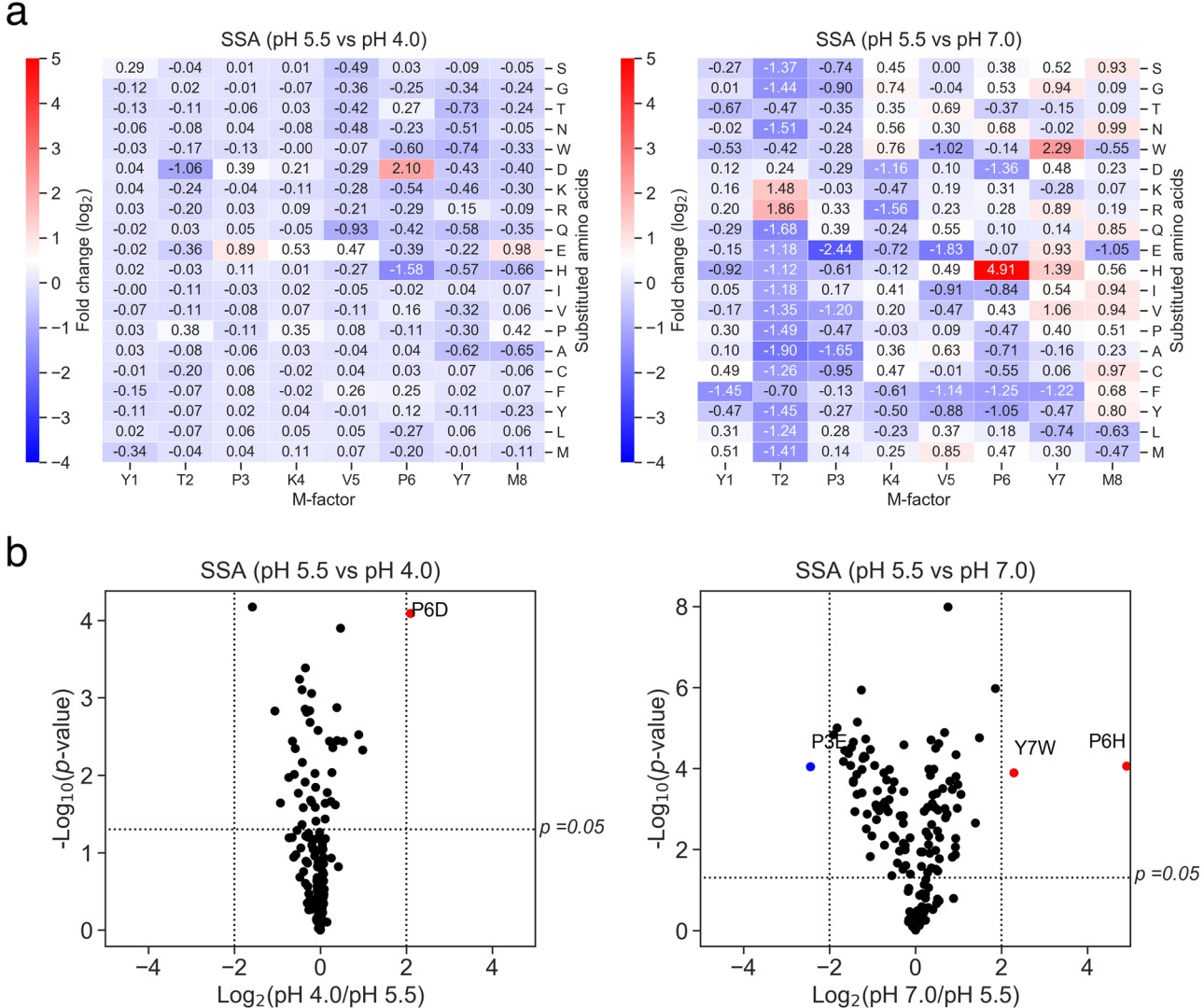

**Fig. 2 | Environmental pH modulates fitness outcomes of M-factor variants.**
**a** Heatmap comparing variant frequencies after five cycles on SSA medium at pH 4.0 and 7.0, normalized to pH 5.5. Log$_2$ fold changes are the mean of three replicates ($n = 3$). P6D was strongly enriched at pH 4.0, whereas P6H and Y7W were enriched at pH 7.0 (log$_2 > 2$). **b** Volcano plot of magnitude vs. statistical significance for (**a**). Variants with ≥ 4-fold enrichment or depletion and adjusted $p < 0.05$ are highlighted in red (enriched) or blue (depleted). P6D (pH 4.0), P6H and Y7W (pH 7.0) met both criteria.

To further investigate the molecular basis of pH sensitivity, a *map4-lacZ* reporter assay was performed using synthetic M-factor peptides. The P6H peptide showed greater than a five-fold increase in β-galactosidase activity at pH 7.0 compared with pH 5.5, indicating enhanced receptor activation under near-neutral conditions. In contrast, the T2Q peptide, included here as a reference variant, exhibited the opposite trend, with reduced signaling at pH 7.0 (Fig. 3c). These results suggested that different pH-responsive variants possess distinct structural or electrostatic features that modulate receptor engagement in a context-dependent manner.

To explore how environmental pH influences mating partner selection, pairwise competition was performed between wild-type and P6H strains on SSA plates across pH 5.5–9.0. Although wild type outcompeted P6H at acidic pH, the relative representation of P6H progressively increased at higher pH levels, rising from ~21% at pH 5.5 to ~51% at pH 8.0, achieving parity at pH 7.5 and modest dominance at pH 8.0 (Fig. 3d). Although the trend did not reach statistical significance, the convergence of fitness across higher pH conditions highlighted the context-dependence of mating ability in this variant. The findings emphasized that P6H, consistently outcompeted under standard laboratory conditions (pH 5.5), can nonetheless match or exceed wild-type fitness under modestly altered environmental conditions.

To test whether such environmental responsiveness persists under more naturalistic conditions, sporulation via P6H was examined on nutrient-rich juice-based media derived from grape, orange, apple, and mixed vegetables. These media differ from standard laboratory formulations in both composition and pH, potentially mimicking ecological niches where wild yeasts engage in mating. After 48 h of incubation, P6H showed detectable sporulation on apple juice and vegetable juice agar (1.6–9.8% of cells; Supplementary Fig. 9), both of which exhibited near-neutral pH values (~6.5–7.0). Although sporulation levels were modest, they were consistent with the enhanced mating efficiency of P6H observed on synthetic medium at pH 7.0. These findings suggested that certain "latent" variants, deemed non-functional under standard laboratory conditions, retain mating competence in environmental contexts that more closely resemble natural habitats.

To examine whether the pH-responsive behavior of P6H was altered or masked in the presence of redundant wild-type M-factor genes, we introduced either wild-type *mfm1⁺* or *mfm1-P36H* into an *h⁹⁰* background containing intact *mfm2⁺* and *mfm3⁺* genes. Mating efficiency of the strain carrying *mfm1-P36H* was modestly lower than that of the strain carrying three wild-type *mfm* genes at pH 5.5, but was indistinguishable from that of

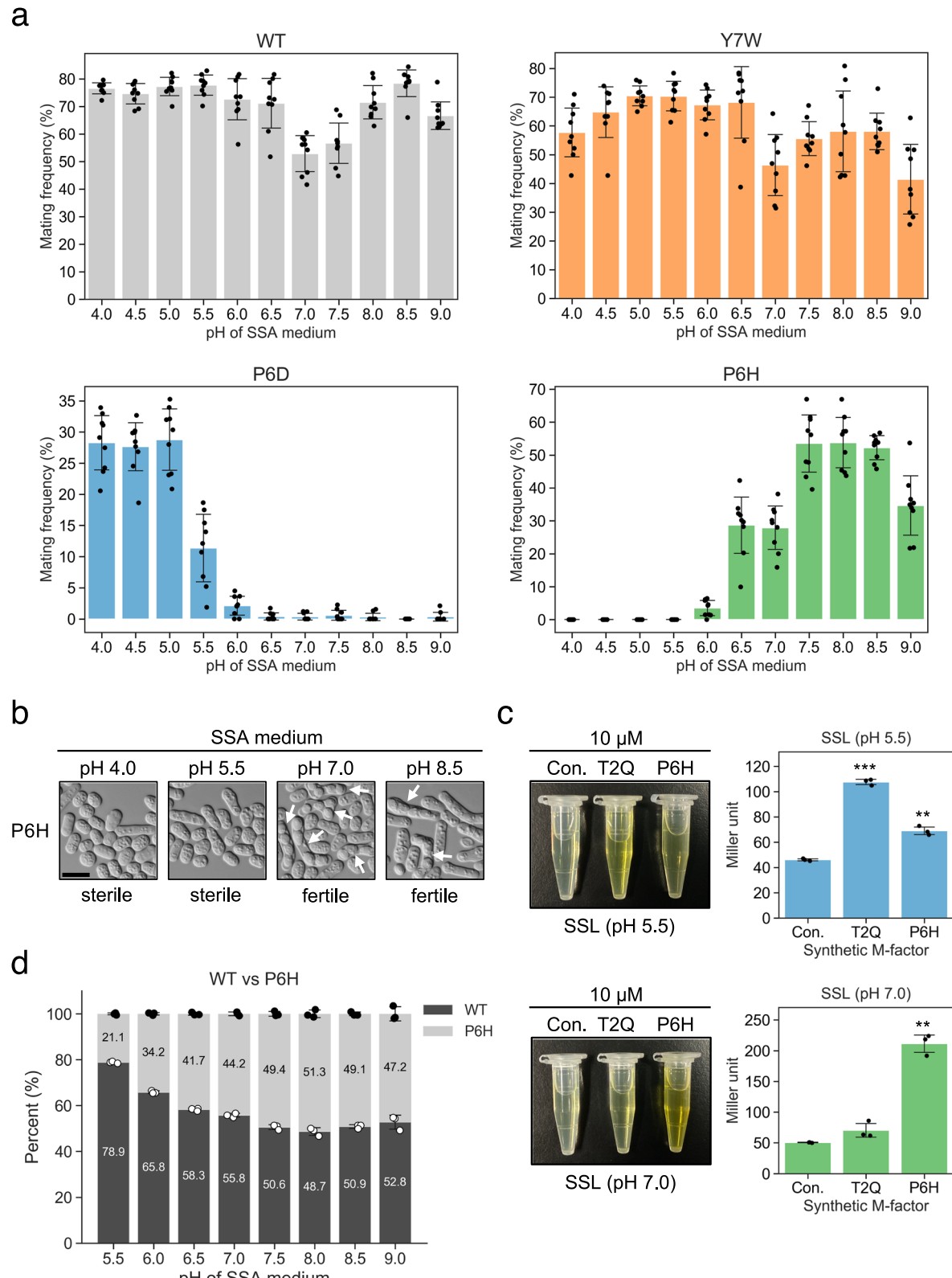

the strain lacking *mfm1* (TS307) on SSA at both pH 5.5 and pH 7.0 (Supplementary Fig. 10). These results indicated that P6H does not exert a dominant-negative effect on mating and that its pH-responsive activity is largely buffered when overall pheromone signaling is sufficient to support mating.

## Context-dependent trade-offs shape the fitness landscape of T2 variants

Motivated by the presence of a T2Q substitution in the M-factor of the closely related species *S. octosporus*[20], we next examined how substitutions at the T2 position affect mating-related fitness in *S. pombe*. In contrast to the

**Fig. 3 | pH-dependent modulation of mating efficiency and receptor activation via M-factor variants. a** Mating frequency of WT, P6D, P6H, and Y7W strains on SSA medium across a pH gradient (4.0–9.0, 0.5-unit increments). After 48 h, mating frequency was calculated from nine random microscopic fields per replicate ($n = 3$). Each dot represents one field; bars are means ±s.d. P6D showed elevated mating at pH 4.0, whereas P6H and Y7W were enhanced at higher pH. **b** Representative differential interference contrast (DIC) images of P6H at four pH values (4.0, 5.5, 7.0, 8.5). Arrowheads indicate mature asci, observed at pH 7.0 and 8.5, but absent under acidic conditions. Scale bar: 5 µm. **c** β-galactosidase assay of Map3 receptor activation via synthetic T2Q and P6H peptides (10 µM) in a *map4-lacZ* reporter strain in nitrogen-free SSL medium at pH 5.5 or 7.0. MeOH served as the control. Bars are means ±s.d. ($n = 3$). P6H activity was >5-fold higher at pH 7.0 vs. pH 5.5; T2Q showed higher activity at acidic pH. Welch's *t*-test comparing each peptide to the MeOH control: pH 5.5, **$p = 0.0054$, ***$p = 1.0 \times 10^{-4}$; pH 7.0, **$p = 0.0038$. **d** Pairwise competition between WT and P6H on SSA medium across a pH gradient (5.5–9.0). Equal numbers were subjected to one mating–sporulation–germination cycle. Relative frequencies were quantified using amplicon sequencing. Bars are means ±s.d. ($n = 3$). WT dominated under acidic conditions, whereas P6H benefited at higher pH, approaching parity with WT at pH 8.0. Welch's *t* test comparing WT and P6H at each pH condition (pH 5.5: $p = 1.0 \times 10^{-8}$; pH 6.0: $p = 2.5 \times 10^{-7}$; pH 6.5: $p = 4.3 \times 10^{-6}$; pH 7.0: $p = 7.3 \times 10^{-5}$; pH 7.5: $p = 0.23$; pH 8.0: $p = 0.12$; pH 8.5: $p = 0.062$; pH 9.0: $p = 0.090$).

**Fig. 4 | Environment-dependent differences in T2 variant frequencies reflect changes in mating efficiency. a** Scatter plot comparing the log$_2$ fold change of 19 T2-substituted M-factor variants after five competition cycles on nutrient-rich YEA medium (x-axis) vs. mating-inducing MEA (y-axis). Points are the mean of three biological replicates ($n = 3$), with error bars indicating s.d. Most variants decreased in frequency on YEA and increased on MEA, consistent with a trade-off between vegetative fitness and mating success. **b** Mating frequency of WT and five selected T2 variants (T2A, T2C, T2D, T2K, T2Q) after incubation for 3 days on YEA at 30 °C. Frequencies were quantified microscopically from nine random fields per strain. Dots represent individual fields; bars show means ±s.d. All variants exhibited significantly higher mating rates than WT under non-inducing conditions (Welch's *t* test: *$p < 0.05$; ***$p < 0.001$).

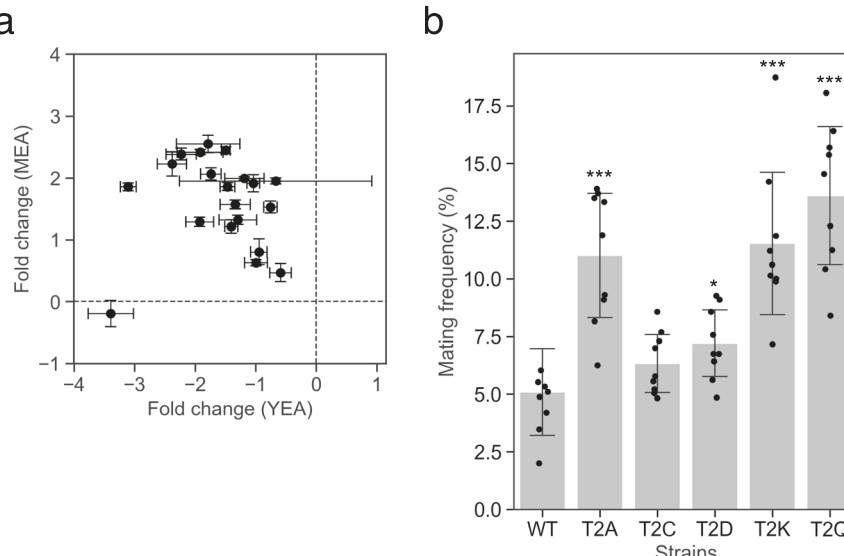

environmentally gated behavior observed at P6, substitutions at the T2 position revealed a distinct form of pleiotropy characterized by enhanced mating efficiency, coupled with reduced vegetative fitness. Variants with mutations in the second M-factor residue (T2), consistently increased in frequency under mating-permissive conditions on MEA (Fig. 1b), suggesting enhanced pheromone signaling under standard laboratory mating conditions. In contrast, the same variants declined under non-mating conditions, such as vegetative growth on rich medium (YEA) (Fig. 1c), indicating an environment-dependent fitness trade-off (Fig. 4a).

This pattern aligns with *S. pombe* physiology, where M-factor activates the mating MAPK pathway and induces G1 cell cycle arrest[24,25]—a metabolically costly state incompatible with rapid mitotic growth, suggesting that T2 substitutions alter the balance between signaling activation and vegetative growth, rather than environmental responsiveness. Sustained or inappropriate activation of this pathway could impair vegetative proliferation. To directly test whether this trade-off can be identified independently of population-level competition, we quantified the mating efficiency and vegetative growth parameters of the wild-type (WT) and T2Q strains under single-strain conditions. Consistent with the competition assays, T2Q showed higher mating efficiency than the WT on MEA, confirming increased mating competence in a mating-permissive environment (Supplementary Fig. 11a). In contrast, during vegetative growth in rich medium (YEL), WT and T2Q reached comparable final OD$_{600}$ values (Supplementary Fig. 11b), however, T2Q exhibited a prolonged lag phase, resulting in delayed population expansion relative to WT (Supplementary Fig. 11c). This growth delay provides a mechanistic explanation for the depletion of the T2Q variant observed under vegetative conditions in the pooled competition assays.

Supporting this, three of five tested T2 variants, including T2Q, exhibited elevated mating frequencies (>10%, ~2-fold ≥WT) on YEA,

where mating is normally repressed (Fig. 4b), implying partial derepression of the sexual program under nutrient-rich conditions. Such derepression likely compromises growth competitiveness, even in the absence of clear growth-rate defects, consistent with partial activation of the mating program causing transient G1 arrest in rich media.

These results illustrated antagonistic pleiotropy, where a single mutation benefits mating-promoting environments and hinders vegetative growth. Such variants may persist in natural populations owing to spatial or temporal variation in mating opportunities or life-history strategies prioritizing reproduction over somatic maintenance.

## Epistatic buffering enables divergent M-factor sequences between species

To investigate how evolutionary divergence in peptide pheromones may be accommodated without loss of function, the M-factor sequences of *S. pombe* and its close relative *S. octosporus* were compared. The *S. octosporus* M-factor differs by three amino acids—T2Q, V5P, and Y7A (Fig. 5a)—and does not activate the *S. pombe* receptor Map3, consistent with previously reported interspecies incompatibility[20,26].

To assess whether these substitutions interact epistatically, the substitutions were introduced individually and in combination into the *S. pombe* M-factor sequence. T2Q alone enhanced mating efficiency, and combined with V5P, it partially restored signaling activity to ~70% of wild-type, indicating that T2Q is a permissive substitution capable of "rescuing" otherwise deleterious changes (Fig. 5b). Notably, these mating assays were conducted under the same mating-permissive conditions used to quantify the T2Q fitness trade-off (Supplementary Fig. 11a), enabling direct comparison between antagonistic pleiotropy and epistatic buffering within a unified experimental framework.

**Fig. 5 | Evolutionary buffering with a permissive mutation expands mutational tolerance in M-factor. a** Sequence alignment of mature M-factor peptides from *S. pombe* and *S. octosporus*, differing at three positions: T2Q, V5P, and Y7A. **b** Mating frequencies of *S. pombe* strains carrying single (T2Q, V5P, Y7A), double (T2Q/V5P, V5P/Y7A, T2Q/Y7A), and triple (T2Q/V5P/Y7A) substitutions. Cells were incubated on MEA at 30 °C for 2 days; frequencies were quantified as in Fig. 4b. Bars show means ±s.d. ($n = 3$). Statistical significance was determined using Welch's *t* test. Comparisons were performed between WT and T2Q (*$p = 0.042$), V5P and T2Q/V5P (***$p = 4.3 \times 10^{-11}$), and Y7A and T2Q/Y7A (**$p = 0.0055$). **c** Rescue of loss-of-function variants via the permissive T2Q mutation. Double mutants combining T2Q with deleterious substitutions at V5 (e.g., V5D, V5H) or Y7 (e.g., Y7A, Y7E) were assayed as in (**b**). Several T2Q-containing double mutants showed partial mating ability recovery, indicating a buffering effect.

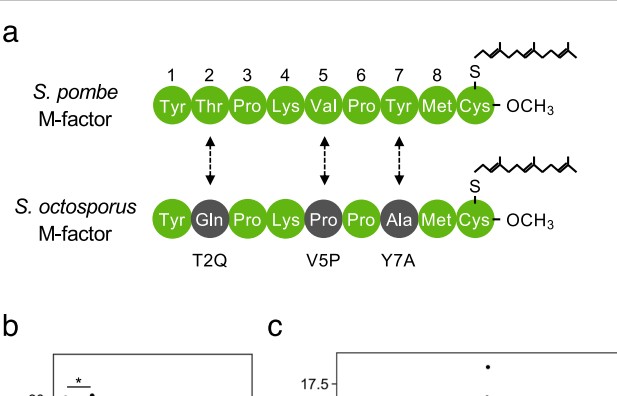

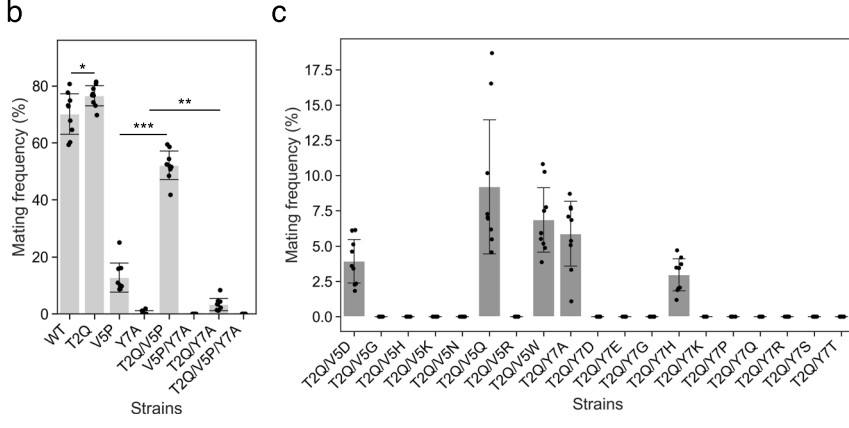

We further examined T2Q's buffering role by introducing it into backgrounds carrying eight V5 and eleven Y7 distinct loss-of-function variants. In several of these double mutants, mating capacity was restored from near-zero to 3–9% (Fig. 5c), showing that T2Q increases the mutational robustness of the pheromone system, broadening the range of sequence changes that can be tolerated without complete functional loss.

Such permissive mutations are well known in molecular evolution as "enablers" that open otherwise inaccessible evolutionary routes, particularly in ligand–receptor systems[27–29]. Although precise mechanisms, such as altered peptide abundance, stability, or receptor interaction, remain to be elucidated, our findings showed that epistatic interactions can preserve signal function without passing through fully non-functional states. Collectively, these results suggested that M-factor diversification is shaped by a combination of mutational constraints, environmental responsiveness, and intramolecular epistasis, with substitutions such as T2Q acting as evolutionary footholds maintaining pheromone system function across a rugged and changing fitness landscape.

## Discussion

Together, these results revealed that the M-factor peptide contains functionally distinct positions that contribute to mating fitness through different mechanisms. While the P6 position acts as an environmentally gated site for which activity is modulated by extracellular pH, the T2 position influences the signaling output through antagonistic pleiotropy, enhancing mating at the expense of vegetative growth without conferring environmental sensitivity. This study demonstrates that the M-factor pheromone in *Schizosaccharomyces pombe*, despite being a minimalist nine-amino-acid peptide, exhibits both environmental responsiveness and evolutionary robustness. High-throughput competition assays, environmental manipulation, and mutational reconstruction showed that M-factor variation can yield context-dependent mating fitness gains while buffering deleterious mutations. These findings illustrated how simple signaling molecules can evolve under complex selective regimes shaped by ecological variability and receptor constraints.

A key finding is that the functional performance of specific M-factor variants depends strongly on environmental conditions, especially extracellular pH. For example, the P6H variant showed minimal activity under

standard laboratory pH; however, it was strongly activated under higher pH conditions (Fig. 2), promoting mating and sporulation. Such context-dependent activation exemplified cryptic functional diversity, where variants neutral or deleterious under one condition became advantageous in another. These phenotypes may be relevant in heterogenous natural habitats. For example, insects, major yeast dispersers, often possess alkaline gastrointestinal tracts[30], which could potentially favor variants such as P6H. Although pH was a major driver, media and juices of similar pH showed differences, indicating additional modulators such as nutrient composition or osmotic stress. Integrating profiling with synthetic pheromone assays could help disentangle these effects.

Variants at the second residue, particularly T2Q, showed strong mating performance; however, they were consistently disfavored in vegetative growth, suggesting antagonistic pleiotropy. M-factor production induces G1 arrest and mating pathway activation, processes that are costly and incompatible with proliferation. Our data indicated that T2Q may partially bypass environmental gating, triggering sporulation even in rich media, reducing mitotic competitiveness. Consistent with this interpretation, pooled competition assays across mating-permissive and non-mating conditions showed that the T2Q was more strongly depleted than the WT during vegetative growth, but more strongly enriched during mating, with these differences increasing over repeated mating–germination cycles. Using $h^{90}$ strains, which alternate between pheromone-producing and -receiving states, ensured all genotypes shared the same receptor environment. This design enabled direct competition among M-factor variants while minimizing receptor-side confounds, such that dynamics reflected integrated signal efficacy across mating, proliferation, and sporulation.

Unpredictably, T2Q partially restored mating in otherwise incompatible variants, including those with deleterious V5 and Y7 mutations. Such permissive mutations, common in enzymes and transcription factors, are rarely documented in secreted peptide signals. T2Q appeared to expand mutational tolerance, enabling evolutionary trajectories otherwise blocked by intermediate loss-of-function. Gene redundancy may further promote such evolvability, as *S. pombe* and *S. octosporus* each possess multiple M-factor genes encoding identical peptides[20,31], creating "safe zones" for variation. We propose that divergence observed in *S. octosporus* M-factor (T2Q, V5P, Y7A) arose through such buffered pathways, with T2Q

appearing early to permit compensatory evolution at other sites. Consistent with this view, introduction of P6H into a wild-type background retaining multiple M-factor genes did not measurably affect mating efficiency relative to strains carrying only wild-type *mfm* genes at either pH tested, indicating that P6H does not exert a dominant-negative effect in a redundant genetic context. This suggests that gene redundancy can mask context-dependent gains in signaling activity when the overall pheromone output exceeds the threshold required for mating, allowing environmentally responsive variants to persist without overt phenotypic consequences. Together, these observations support the idea that redundancy can permit the retention of cryptic variants and buffer their phenotypic effects until the ecological or genetic contexts render them visible for selection.

The assays in this study measured relative fitness under laboratory conditions, which may not fully capture natural selective pressures. The mating efficiency was inferred from bulk dynamics and sporulation, thereby integrating multiple processes including pheromone production, secretion, stability, and receptor activation. Consequently, certain mutations may occur at upstream steps, such as expression or processing, rather than peptide–receptor binding. Moreover, the influence of buffering and permissive effects is based on phenotypic restoration and competitive outcomes, rather than on direct structural or biophysical measurements of peptide–receptor interactions. Although our genetic, physiological, and reporter assays established robust pH-dependent activation of the P6H variant, the molecular mechanism underlying this environmental sensitivity remains unresolved. Our plausible explanation is that protonation of histidine at position 6 alters local electrostatics, peptide conformation, or receptor contact geometry, thereby modulating binding affinity or receptor activation thresholds in a pH-dependent manner. Such effects are difficult to disentangle using genetic assays alone, particularly for short, flexible, lipid-modified peptides that act on GPCRs. Resolving these mechanisms requires complementary approaches, including mass spectrometry to assess peptide abundance and processing, quantitative receptor-binding assays or signaling assays using synthetic peptides, and high-resolution structural and computational analyses. Particularly, membrane-embedded molecular dynamics or constant-pH simulations of the Map3–M-factor complex could provide valuable insights into how protonation states influence peptide–receptor interactions across physiological pH ranges. Although such analyses are beyond the scope of this study, they represent an important direction for future studies aimed at linking environmental responsiveness to molecular mechanisms.

Natural *S. pombe* isolates show complete conservation of M-factor and its receptor Map3, while polymorphism accumulates primarily in P-factor and Mam2[20]. This suggests asymmetric constraint, with one axis serving as a compatibility gatekeeper and the other enabling adaptive flexibility. This study identified permissive mutations such as T2Q, which supports the idea that subtle structural changes can expand functional space without disrupting reproductive integrity. Sex-specific ligand–receptor system differences in yeasts[22,32,33], amphibians[34,35], and insects[36,37] illustrate how environmental heterogeneity and structural constraints shape the balance between specificity and evolvability[12,16,38–40]. Overall, the M-factor system of *S. pombe* offers a tractable model for understanding how small peptide signals balance environmental responsiveness, structural constraint, and evolutionary potential. By revealing how cryptic alleles, trade-offs, and buffering mutations shape sexual communication, this work provides a framework for studying peptide signaling in dynamic ecosystems. Notably, none of the ~150 *S. pombe* wild isolates examined to date carry amino acid substitutions in M-factor, despite having multiple identical gene copies[20]. Broader ecological sampling may reveal hidden diversity, as no insect-derived isolates have yet been analyzed and such isolates are rare worldwide[20,41,42]. To our knowledge, pheromone systems of insect-derived *S. pombe* have never been investigated, even though the species has occasionally been reported from bee-associated substrates such as honey or dried fruits[43]. Targeting insect-derived strains could uncover novel alleles. Moreover, related fission yeasts such as *Schizosaccharomyces japonicus*, abundant in Japan[44], offer opportunities to assess whether similar constraints or diversification occur in other *Schizosaccharomyces* species. Extending this framework to more distantly related yeasts will clarify whether the observed patterns represent a general feature of fungal pheromone evolution.

## Methods

### Strains, media, and culture conditions

*Schizosaccharomyces pombe* strains used in this study are listed in Supplementary Data 1. Standard media included YEL (Yeast Extract Liquid: 0.5% yeast extract, 3% D-glucose) and YEA (YEL with 1.5% agar). Mating and sporulation were induced on MEA (Malt Extract Agar: 3% malt extract, 1.5% agar, pH 5.5). Additional media used included EMM2 (Edinburgh Minimal Medium 2: EMM2 base with 1/5 nitrogen concentration: 1 g/L $NH_4Cl$, pH 5.5)[45], PMG (Pombe Minimal Glutamate: EMM2 with 3.75 g/L L-glutamic acid replacing $NH_4Cl$)[46], and SSA (Synthetic Sporulation Agar: pH 5.5)[24]. For ecological assays, semi-natural media were prepared using juice and agar: grape, orange, and apple juice (each at 50%), or vegetative juice (20% V8)[47], supplemented with 0.2% $CaCO_3$ and solidified with agar (2% for fruit juices, 1.5% for vegetable juice).

Juice products were sourced from commercial suppliers: Welch's 100% grape juice (Kraft Heinz), Dole 100% orange and apple juice (Dole Japan), and V8 vegetable juice (Campbell Japan). The pH of each medium was measured before use: grape (pH 5.7), orange (pH 4.7), apple (pH 6.5), and vegetable (pH 6.2).

Supplemental stocks were prepared as follows. 50× salts stock contained 5.25 g $MgCl_2·6H_2O$, 0.0735 g $CaCl_2·2H_2O$, 5.0 g KCl, and 0.2 g $Na_2SO_4$ per 100 mL. The 1000× vitamin stock included 0.1 g pantothenic acid, 1.0 g nicotinic acid, 1.0 g inositol, and 0.001 g biotin per 100 mL. The 10,000× mineral stock comprised 0.5 g boric acid, 0.4 g $MnSO_4$, 0.4 g $ZnSO_4·7H_2O$, 0.2 g $FeCl_2·6H_2O$, 0.04 g molybdic acid, 0.1 g potassium iodide, 0.04 g $CuSO_4·5H_2O$, and 1.0 g citric acid per 100 mL of water[48].

Unless otherwise specified, *S. pombe* cultures were incubated at 30 °C.

*Note*: Although derived from the EMM2 formulation, 1 g/L $NH_4Cl$ (one-fifth the standard concentration) was consistently used to enhance sporulation. For simplicity, this modified medium is referred to as "EMM2" throughout the manuscript.

### Competition experiments

In total, 153 *S. pombe* strains, composed of M-factor mutants[11] and wild type, were cultured for 20 h on YEA plates at 30 °C. All strains were in the $h^{90}$ background, which undergoes spontaneous mating-type switching between M- and P-types[49]. This switching ensures each genotype alternates between signal production (M-type) and reception (P-type) during the assay, such that selection specifically targets the M-type state where the M-factor sequence determines mating success.

Following cultivation, colonies were suspended in sterilized water, transferred to 1.5 mL tubes (Eppendorf, Hamburg, Germany) and $OD_{600}$ was measured and adjusted to equal cell densities before mixing. Equal-volume mixtures of all 153 strains were independently prepared in three 50 mL tubes (Corning Inc., Corning, NY, USA) and stored as glycerol stocks (designated TS742–TS744). From a suspension adjusted to $OD_{600} = 30$, 1 mL was collected for genomic DNA extraction using the Wizard Genomic DNA Purification Kit (Promega, Madison, WI, USA). Before extraction, cells were treated with 293 μL 50 mM EDTA and 7.5 μL Zymolyase (1000 U; Zymo Research, Irvine, CA, USA) at 30 °C for 1 h in storage buffer (500 μL), and subsequently processed according to the manufacturer's instructions.

A standard competition cycle involved spotting 100 μL of suspension ($OD_{600} = 10$; ~$2 \times 10^8$ cells/mL) onto assay plates and incubating at 30 °C. Every 4 days, all cells were recovered, resuspended in sterilized water, and treated with 30% ethanol (Wako Pure Chemical Industries, Ltd., Osaka, Japan) for 30 min to remove vegetative cells[21]. Ethanol treatment was omitted for cultures on YEA plates. From the second cycle onward, inoculation volumes were 100 μL at $OD_{600} = 20$ for all media except YEA ($OD_{600} = 10$).

Genomic DNA for sequencing was obtained by inoculating recovered cells in 10 mL YEL at an initial OD$_{600}$ of 0.2 and incubating for 2 days at 30 °C. Competitions were conducted on YEA, MEA, and EMM2 at pH 4.0, 5.5, 7.0 and at elevated temperature (35 °C), and on SSA at pH 4.0, 5.5, and 7.0. Medium pH was adjusted using HCl or NaOH.

For clarity, *cycle 1* refers to the initial competition round in which all strains start at equal abundance, and *cycle 5* represents the strain composition after five successive competition cycles under the same environmental condition.

### Quantitation of amplicon-based pheromone ORF barcodes via MiSeq

Genomic DNA was used as a template for PCR amplification with KOD FX Neo (Toyobo, Osaka, Japan) using primers oTS942/oTS943 (Supplementary Data 2) for 25 cycles. PCR products were purified using the QIAquick PCR Purification Kit (Qiagen, Hilden, Germany), and 1 ng was used for MiSeq library preparation. Libraries were indexed using the Nextera XT Index Kit v2 Set A (Illumina, San Diego, CA, USA), followed by a 12-cycle PCR (annealing temperature 55 °C). Double-stranded DNA concentrations were determined using a Qubit 3 Fluorometer (Thermo Fisher Scientific, Waltham, MA, USA). Mixed libraries were diluted to 4 nM, denatured with NaOH, and sequenced on a MiSeq platform using the MiSeq Reagent Kit v3 (150 cycles; Illumina).

The 75 bp reads encompassed the mature M-factor coding region within *mfm1*, enabling sequence identification and quantification. Reads containing the AAGAAT motif immediately upstream of the mature M-factor coding sequence were extracted, and the subsequent 24 bp sequences were retrieved using SeqKit[50]. These 24 bp barcodes corresponded to 153 unique genotypes (Supplementary Data 1). Read counts for each genotype were determined at each time point, normalized to the initial read counts, and converted to relative cell abundance. Each sample yielded at least 400,000 reads. Raw read counts are provided in Supplementary Data 3.

### Frequency of zygote formation

Cells were pre-cultured on YEA medium at 30 °C overnight, washed with 1 mL sterilized water (10,000 rpm, 3 min, 4 °C), and adjusted to an OD$_{600}$ of 5.0. Aliquots (30 µL) were spotted on YEA, MEA, SSA, or fruit/vegetable juice media (grape, orange, apple, vegetable) and incubated at 30 °C for 48 h. After 48 h, cells were harvested and examined using a differential interference contrast microscopy (BX53, Olympus, Tokyo, Japan) with the UPlanSApo 60×/1.35 NA oil-immersion objective, ORCA-spark camera (Hamamatsu Photonics, Hamamatsu, Japan), and CellSens software. Nine random fields were imaged per plate. Mating rates were calculated as in Seike et al.[11] using the equation:

$$\text{Mating rate}(\%) = \left\{(2Z + 2A + 0.5S) \times 100\right\}/(V + 2Z + 2A + 0.5S)$$

where $V$, $Z$, $A$, and $S$ represent vegetative cells, zygotes, asci, and spores, respectively. Data are reported as the mean ±s.d. Raw counts are provided in Supplementary Data 4.

### Growth assays

Growth assays were performed to quantify the vegetative fitness under nutrient-rich conditions. Cells were precultured overnight on YEA plates at 30 °C, harvested in sterilized water, and inoculated into YEL medium at an initial optical density (OD$_{600}$) of 0.1. For each strain, 200 µL of cell suspension was dispensed into individual wells of CELLSTAR 96-well flat-bottom plates (Greiner Bio-One, Kremsmünster, Austria).

Cells were cultivated for 24 h at 30 °C with continuous orbital shaking, and OD$_{600}$ was measured automatically every 10 min using an Infinite 200 PRO microplate reader (Tecan, Männedorf, Switzerland). Final OD$_{600}$ values were recorded at the end of the incubation period. To quantify differences in lag phase, the time required to reach defined OD$_{600}$ thresholds (OD$_{600}$ = 0.5) was additionally calculated from growth curves.

Each assay was performed with four biological replicates ($n = 4$) per strain. Data are reported as the mean ±s.d. Raw counts are provided in Supplementary Data 5.

### In vitro M-factor activity assay using a *map4* promoter–*lacZ* fusion

β-galactosidase activity was measured to evaluate activation of the M-factor receptor Map3 using synthetic M-factor peptides, following a previously described method[22]. Tester strain TS472 was inoculated into 5 mL SSL + N medium (SSA without agar) at an initial OD$_{600}$ of 0.1 and cultured at 30 °C for 20 h. Cells were washed three times with SSL−N medium (lacking 0.5 g/L aspartic acid) and resuspended in SSL−N at varying pH values (adjusted with HCl or NaOH) to OD$_{600}$ = 2.0. Synthetic M-factor (Eurofins) was added to a final concentration of 10 µM, and cultures were incubated at 30 °C for 24 h.

Thereafter, cells were washed and resuspended in Z-buffer (60 mM Na$_2$HPO$_4$, 40 mM NaH$_2$PO$_4$, 10 mM KCl, 1 mM MgSO$_4$·7H$_2$O, and 50 mM β-mercaptoethanol; pH 7.0) to OD$_{600}$ = 5.0. For each replicate ($n = 3$), 200 µL of cell suspension was mixed with 600 µL Z-buffer, 15 µL chloroform, and 12 µL 0.1% SDS, vortexed, and incubated at 30 °C for 5 min. Reactions were initiated with 160 µL *o*-nitrophenyl-β-D-galacto-pyranoside (ONPG; 4 mg/mL in Z-buffer), and both start and end times were recorded. After incubation at 30 °C, reactions were terminated with 400 µL 1 M Na$_2$CO$_3$, centrifuged (13,000 rpm, 3 min, 4 °C), and 200 µL supernatant was diluted into 1 mL 0.33 M Na$_2$CO$_3$. Absorbance at 420 nm (OD$_{420}$) was measured to calculate Miller units.

$$\text{Miller units} = (\text{OD}_{420} \times 1000)/(V \times t \times \text{OD}_{600})$$

where $V$ is reaction volume (mL) and $t$ is reaction time (minute). Data are reported as the mean ± s.d.

### Construction of plasmids and strains

Primers and plasmids are listed in Supplementary Data 2, 6, respectively. All PCRs were performed with KOD FX Neo DNA polymerase. Plasmids pTS426–pTS433, carrying *mfm1* alleles T32Q/V35D, T32Q/V35H, T32Q/V35K, T32Q/V35R, T32Q/V35N, T32Q/V35Q, T32Q/V35G, and T32Q/V35W, respectively, were generated via inverse PCR from the template plasmid pTS369 (*mfm1*-T32Q)[11] using primer pairs oTS1062/oTS1061, oTS1063/oTS1061, oTS1064/oTS1061, oTS1065/oTS1061, oTS1066/oTS1061, oTS1067/oTS1061, oTS1068/oTS1061, and oTS1069/oTS1061.

Plasmids pTS434–pTS444, carrying *mfm1* alleles T32Q/Y37D, T32Q/Y37E, T32Q/Y37H, T32Q/Y37K, T32Q/Y37R, T32Q/Y37Q, T32Q/Y37S, T32Q/Y37T, T32Q/Y37A, T32Q/Y37G, and T32Q/Y37P, respectively, were generated via inverse PCR from the template plasmid pTS369 using primer pairs oTS1071/oTS1070, oTS1072/oTS1070, oTS1073/oTS1070, oTS1074/oTS1070, oTS1075/oTS1070, oTS1076/oTS1070, oTS1077/oTS1070, oTS1078/oTS1070, oTS1079/oTS1070, oTS1080/oTS1070, and oTS1081/oTS1070.

PCR products were digested with DpnI (New England Biolabs, Ipswich, MA, USA) to remove template DNA, phosphorylated with T4 polynucleotide kinase (TaKaRa, Shiga, Japan), self-ligated using T4 DNA ligase (TaKaRa), and transformed into *Escherichia coli* DH5α competent cells (New England Biolabs). Mutations were verified via Sangar sequencing (Eurofins). Each plasmid was linearized with BamHI within the *ade6* selection marker and integrated into recipient *S. pombe* strains via electroporation[11]. Transformants were selected on SD plates, and integrated *mfm1* alleles were confirmed using colony PCR and sequencing.

To generate strains carrying the P6H variant in a wild-type M-factor background, strain TS307, which retained intact *mfm2+* and *mfm3+* genes, was constructed by crossing FY23412 and FY7120. To introduce the P6H substitution, plasmid pTS814 (*mfm1-P36H*) was generated by inverse PCR using pTS362 (*mfm1+*) as a template and the primer pair oTS1700/oTS1701. Plasmids pTS362 and pTS814 were linearized at the BamHI site within the *ade6* locus and integrated into TS307, as described above. Transformants

were selected on SD + Ura (200 mg/L) plates, and correct integration was confirmed using colony PCR and sequencing, yielding strains TS1373 and TS1374.

## Statistics and reproducibility

All statistical analyses treated biological replicates ($n = 3$) as the unit of replication. Unless stated otherwise, values are reported as mean ± s.d., tests were two-sided, and no data were excluded. For two-group comparisons we used Welch's $t$ test (unequal variances). Significance in figures is denoted as $p < 0.05$, $p < 0.01$, and $p < 0.001$. For amplicon-based competition assays, reads mapping to the 153 genotypes were normalized by library size to obtain variant frequencies. Selection was summarized as $\log_2$ fold-change ($\log_2$FC) within each replicate: $\log_2\text{FC} = \log_2(\text{frequency}_{cycle}/\text{frequency}_{start})$. pH effects were tested versus the reference (pH 5.5). Volcano plots highlight variants with $|\log_2\text{FC}| \geq 2$. For microscopy-based mating assays, nine random fields were scored per plate; field values were averaged within each biological replicate to avoid pseudoreplication. For β-galactosidase (map4–lacZ) assays, Miller units were computed per biological replicate. The primary tests compared each peptide condition (10 μM) against the matched vehicle control (MeOH) at the same pH. Reproducibility was assessed by examining correlations among biological replicates and between observed post-competition frequencies and predictions from single-strain mating efficiencies were assessed using Pearson's r with $R^2$ reported. Analyses were performed in Python (pandas, numpy, math); plots were generated using matplotlib and seaborn.

## Reporting summary

Further information on research design is available in the Nature Portfolio Reporting Summary linked to this article.

## Data availability

All data generated or analyzed during this study are included in the published article and its Supplementary Information files (Supplementary Data 3–5). Source data for charts/graphs can be found in Supplementary Data 7. Raw amplicon sequencing reads have been deposited in the DDBJ Sequence Read Archive under BioProject PRJDB40561. Strains and plasmids generated in this study have been deposited at the NBRP, Japan. The strains are available under accession numbers FY60341–FY60362, and the plasmids under accession numbers FYP7559–FYP7580. All other data supporting the findings of this study are available from the corresponding author upon reasonable request.

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

## Acknowledgements

We thank the National BioResource Project (NBRP), Japan, for providing yeast strains and plasmids. We also thank the members of the Furusawa lab at RIKEN for their valuable discussions and technical advice throughout the project. This work was supported by JSPS KAKENHI (JP19K16197 and JP20H04790), a Special Postdoctoral Researcher Grant from RIKEN, and JST PRESTO (JPMJPR24N9) to T.S. The funding agencies played no role in study design, data collection and analysis, decision to publish, or manuscript preparation.

## Author contributions

T.S. conceived the study, designed and performed all experiments, analyzed the data, and wrote the manuscript. N.S. constructed selected plasmids and strains and performed mating assays under the supervision of T.S. H.K. prepared sequencing libraries and assisted next-generation sequencing data collection. C.F. provided scientific guidance, supervision, and critical input on manuscript structure and revisions. T.S. secured all funding that supported this research.

## Competing interests

The authors declare no competing interests.
