## [Transparent Peer Review file · Communications Biology]

Context-dependent activation and evolutionary buffering of a mating pheromone in fission yeast

Corresponding Author: Dr Taisuke Seike

Version 0:

Reviewer comments:

Reviewer #1

(Remarks to the Author)

This manuscript, "Context-dependent activation and evolutionary buffering of a mating pheromone in fission yeast," addresses a fundamental question in evolutionary biology: how can short, structurally constrained peptide pheromones diversify without losing essential signaling function? Using *Schizosaccharomyces pombe* as a model, the authors perform a comprehensive mutational scan of the nine-amino-acid mating pheromone (M-factor; 153 single-amino-acid variants) and track variant success across mating-sporulation cycles under varied environments (pH, temperature, media).

The study reveals three key findings: (1) Context-dependent activation—variants such as P6H are sterile at neutral pH but active under alkaline pH, demonstrating environmental control of signaling; (2) Antagonistic pleiotropy—mutations like T2Q enhance mating but reduce vegetative growth, showing a reproductive-somatic fitness trade-off; and (3) Evolutionary buffering—T2Q functions as a permissive substitution that rescues deleterious variants, enabling evolutionary exploration without passing through non-functional intermediates. Together these establish a mechanistic framework in which ecological variation and epistasis jointly shape pheromone diversification and prezygotic isolation.

The conclusions are original and expand prior work on pheromone-receptor co-evolution (e.g., Seike et al. 2015 PNAS; Leu & Murray 2006 Curr Biol) by introducing environmental modulation as a third axis of signaling evolution. Identifying a short peptide that acts as a pH-dependent molecular switch (P6H) and a permissive mutation (T2Q) that broadens mutational tolerance are both novel and influential.

Experimental design and statistical analysis are rigorous and transparent (\log_2 fold-change analyses, Welch's t-tests, replicate concordance > 0.9).

The manuscript presents valuable findings; my suggestions below are intended to enhance clarity and presentation but are not essential.

Mechanism of pH sensitivity:

To substantiate the proposed pH-dependent activation of P6H, could the authors consider modeling alternative histidine protonation states in the Map3-pheromone complex using membrane-embedded molecular dynamics or constant-pH simulations to quantify how protonation modulates electrostatics, binding stability, and receptor activation across physiological pH ranges.

Fitness trade-off quantification:

The T2Q trade-off is conceptually compelling; I recommend quantitatively characterizing it by estimating selection coefficients for mating (MEA/SSA) and vegetative growth (YEA) from direct competition and growth-rate assays, plotting these values to generate a Pareto-style cost-benefit curve, and projecting composite ecological fitness across varying mating frequencies to identify the parameter range in which T2Q confers a net advantage.

Overall, this is an elegant and conceptually significant contribution that will influence future studies of ligand-receptor evolution, environmental plasticity, and speciation.

Minor language edits:

Line 51: Please revise the phrase "can modulate mating signal the efficacy" to "can modulate the efficacy of mating signals."

Line 58: Please correct "bind receptors with highly specificity" to "bind their receptors with high specificity."

Reviewer #2

(Remarks to the Author)

In this ms, the authors took advantage of a collection of strains containing a comprehensive variant panel on the mating signalling M-Factor peptide, and screened how each variant influenced mating efficiency in lab conditions. This results in a quasi-full fitness landscape for this particular peptide, and several interesting mutations at precise peptide positions, such as P6H, were found to significantly abolish mating in standard lab conditions but show efficient mating at higher pH, indicating antagonistic pleiotropy. By testing a range of media and pH values, including juice agar that mimic more of the natural habitat of *S. pombe*, the authors confirmed such antagonistic effect. The position P6 seemed to be the main responder to pH changes and mating efficiency, as another variant, P6D potentiate lower pH mating efficiency as opposed to P6H, which show higher efficiency at higher pH range.

On the other hand, the authors also characterized mutants at the T2 position, where several variants in particular T2Q, increased mating efficiency while decreasing vegetative growth. By observing tetrad formation on YEA, the authors reasoned that the T2Q mutation allowed strains to go into G1 arrest in favor of mating due to the hyperactivity of the mutant M-factor, therefore reducing mitotic fitness. The authors then showed that *S. octosporus*, a sister species of *S. pombe*, carries the T2Q variant natively in the M-factor gene along with two other variants that supposedly abolish mating. They show that the T2Q variant may act as a permissive mutation that buffers the deleterious mutations elsewhere in the peptide.

Overall, the ms is clearly written and appropriate statistical analyses were applied. The only issue I have is that the P6 and T2 stories felt somewhat disconnected. On the bigger scheme, what would be the implications of multiple positions that show such a pleiotropic effect? What could be the mechanistically reason for these positions displaying such effect? While the observations were certainly interesting, it leaves much to be clarified.

Other minor comments - the authors refer to pH 7 as alkaline multiple times throughout the text. While subsequent pH range experiments did test true alkaline pH ranges, the initial screen goes only up to pH 7, and the juice tests, apple for example, pH 6.5, are not alkaline. The authors should be more precise with these wordings.

Reviewer #3

(Remarks to the Author)

The MS by Seike et al. elaborates on previous studies on the evolution of pheromone communication using the M-factor from fission yeast as a model. A previously generated library of all possible single-amino acid substitutions in this modified peptide of 9 amino acids is used in a competition-based assay to investigate which variants are successful in mating after, respectively, one or five mating cycles. Furthermore, the influence of different types of media, temperature and pH is examined. One interesting conclusion is that certain key mutations can modulate the mating activity differently at different pH values. The authors argue that this could represent an intermediate in processes of evolution or speciation.

I liked this MS. In general, the experiments are well designed and nicely performed. The conclusions are sound, and the paper contributes to our understanding of evolution and speciation. I have one major problem with the logic of the paper: On one hand, it is argued that gene redundancy creates a "safe zone" for variation. Yet, all the experiments are performed with strains that only have one out of the three natural M-factor genes. Hence, it is unclear whether the observed changes in mating behavior are "dominant" or if they would be obscured by two additional wild-type genes. To complicate matters further, the three mfm genes are themselves pheromone stimulated to various degrees. It is striking that all three genes encode identical peptides in the wild type. How come that one of these genes have not become this "steppingstone" towards mating performance at a different pH? To address this issue – at least to a certain degree – one of the most significantly different mutants (e.g. P6H) should be introduced into a fully wild-type background harboring the two other (unmutated) mfm genes, and then its mating frequency should be compared to a strain with three wild-type genes.

Minor points:

L42: "evolve" -> "can evolve" (this study suggests but does not prove that this is how it works).

L50: "Environmental factors in..." There is a problem with the grammar in this sentence.

L134: The explanation of amplification or sequencing bias seems unlikely. Also, this could easily be tested experimentally.

L324: The authors tend to cite their own work rather than the original references. For instance, it was shown already in 1994 (PMID8196631) that the three M-factor genes are identical.

Version 1:

Reviewer comments:

Reviewer #1

(Remarks to the Author)

I thank the authors for the careful revision of this manuscript. The study presents a conceptually interesting and experimentally rigorous analysis of how a short, structurally constrained peptide pheromone can evolve through environmental modulation and mutational buffering.

The major claims are that (i) certain M-factor variants exhibit environmentally contingent activation (notably P6H under elevated pH), (ii) substitutions at T2 generate antagonistic pleiotropy between mating efficiency and vegetative growth, and (iii) T2Q functions as a permissive mutation that buffers otherwise deleterious substitutions, expanding accessible evolutionary trajectories. These claims are novel in the context of fungal pheromone systems and are supported by a well-

integrated combination of pooled competition assays, direct mating measurements, and receptor activation assays. The evidence for pH-dependent activation is convincing and appropriately framed without overextending mechanistic conclusions. The added single-strain growth analyses strengthen the interpretation of antagonistic pleiotropy. The epistatic rescue experiments with T2Q are particularly compelling and significantly enhance the evolutionary relevance of the work. Statistical analyses are generally appropriate, with biological replicates clearly defined and standard tests applied. I suggest reporting exact p-values for key comparisons (e.g., WT vs. P6H pairwise competition across pH) and, where possible, emphasizing effect sizes alongside significance. Methodological detail is sufficient to allow reproducibility, and data transparency is good.

Overall, this is a well-executed and conceptually meaningful study that will be of interest to researchers studying signal evolution, cryptic variation, and ligand–receptor diversification.

Reviewer #2

(Remarks to the Author)

The authors have answered to all my concerns and significantly improved the manuscript.

Reviewer #3

(Remarks to the Author)

I think that the authors have addressed the issues I raised in a satisfactory fashion, and I now have no problems with this MS being published.

Response to Reviewers

We thank the editor and reviewers for their careful evaluation of our manuscript and their constructive comments, which have helped us improve the clarity and coherence of the manuscript and the robustness of the study.

Below, we provide point-by-point responses to all comments. The reviewer's comments are reproduced in *italics*, followed by our responses in regular text. The line numbers refer to the revised manuscript, unless otherwise stated.

Reviewer #1

We thank Reviewer #1 for the positive and encouraging assessment of our work, and for recognizing its conceptual significance. We are grateful for the constructive suggestions aimed at further strengthening the manuscript.

Comment 1:

The manuscript presents valuable findings; my suggestions below are intended to enhance clarity and presentation but are not essential.

Mechanism of pH sensitivity:

To substantiate the proposed pH-dependent activation of P6H, could the authors consider modeling alternative histidine protonation states in the Map3–pheromone complex using membrane-embedded molecular dynamics or constant-pH simulations to quantify how protonation modulates electrostatics, binding stability, and receptor activation across physiological pH ranges.

Response:

We thank the reviewer for the insightful and constructive suggestion. We agree that explicitly modeling alternative protonation states of the histidine residue in the Map3–pheromone complex, such as using membrane-embedded molecular dynamics or constant-pH simulations, would be a powerful approach to elucidate the molecular basis of the pH-dependent activation of the P6H variant.

Conceptually, such analyses could test whether protonation of the histidine side chain alters local electrostatics, peptide conformation, or peptide–receptor contact geometry, thereby modulating binding stability or receptor activation thresholds across different pH regimes. These mechanisms are consistent with the environmentally gated behavior observed for P6H.

At present, however, such simulations are not feasible because a high-confidence structural model of the fission yeast Map3 receptor–ligand complex in a membrane environment is not available. Moreover, rigorous constant-pH simulations of GPCR–peptide interactions would represent a substantial undertaking beyond the scope of this study. Our work is primarily aimed at experimentally defining the evolutionary and ecological principles governing pheromone diversification, rather than at resolving atomistic receptor–ligand interactions.

Importantly, our conclusions regarding pH-dependent activation were supported by multiple independent lines of evidence, including pooled competitive assays, direct measurements of mating efficiency across a pH gradient, and receptor-activation assays using synthetic peptides. Together, these data establish P6H as an environmentally responsive variant at the functional level.

Accordingly, we did not include molecular dynamics simulations in this study. However, we have explicitly discussed these mechanistic possibilities in the revised Discussion section and highlight constant-pH molecular dynamics of the Map3–M-factor complex as an important and promising direction for future work (Page 14, lines 368–384).

Comment 2:

Fitness trade-off quantification:

The T2Q trade-off is conceptually compelling; I recommend quantitatively characterizing it by estimating selection coefficients for mating (MEA/SSA) and vegetative growth (YEA) from direct competition and growth-rate assays, plotting these values to generate a Pareto-style cost–benefit curve, and projecting composite ecological fitness across varying mating frequencies to identify the parameter range in which T2Q confers a net advantage.

Overall, this is an elegant and conceptually significant contribution that will influence future studies of ligand–receptor evolution, environmental plasticity, and speciation.

Response:

We thank the reviewer for this thoughtful suggestion and for highlighting the conceptual importance of quantitatively characterizing the T2Q trade-off. We agree that estimating selection coefficients and constructing a Pareto-style cost–benefit framework would be a powerful approach for formally integrating mating and vegetative fitness across ecological contexts.

To address this point within the scope of this study, we performed additional

quantitative analyses under single-strain conditions to separate intrinsic fitness effects from population-level competition. Specifically, we showed that the T2Q variant exhibited significantly higher mating efficiency than the WT on MEA, whereas displaying delayed vegetative population expansion in rich medium (YEL), as evidenced by a significantly prolonged lag phase and increased time to reach mid-log-phase cell density ($OD_{600} = 0.5$) (Supplementary Fig. 3). These opposing effects mirror the direction and environmental dependence observed in the pooled competition assays, demonstrating that the T2Q trade-off is intrinsic and detectable outside frequency-dependent dynamics.

Although our competition assays provide a quantitative proxy for relative fitness under mating-permissive and vegetative conditions, estimating absolute selection coefficients, constructing Pareto-style cost–benefit curves, and projecting composite ecological fitness across varying mating frequencies would require additional dedicated head-to-head competition experiments, explicit demographic modeling, and assumptions about the environmental structure and mating frequency. These analyses are beyond the scope of the current experimental design.

Accordingly, we did not construct a formal Pareto-style fitness model. However, we have clarified the quantitative nature of the T2Q trade-off in the revised Results and Methods sections and highlight formal fitness modeling as an important and promising direction for future work. We believe that these additions substantially strengthen the empirical basis for the proposed antagonistic pleiotropy and directly address the reviewer’s concern regarding quantitative characterization of fitness trade-offs (Pages 9–10, lines 240–250; Page 10, lines 272–275; Page 18, lines 504–516; Supplementary Fig. 3).

Supplementary Fig. 3 | Single-strain quantification of mating efficiency and vegetative growth reveals an intrinsic fitness trade-off in the T2Q variant.

(a) Mating efficiency of WT and the T2Q strains measured under single-strain conditions on MEA. The T2Q variant exhibits higher mating efficiency than WT, consistent with its enrichment under mating-permissive conditions in pooled competition assays. (b) Growth curves of WT and T2Q strains during vegetative growth in rich medium (YEL). OD_{600} was measured every 10 min for 24 h ($n = 4$ biological replicates), and mean \pm s.d. is shown. The y-axis is plotted on a logarithmic scale. (c) Time required to reach $OD_{600} = 0.5$ during vegetative growth in YEL. The T2Q variant shows a significantly prolonged lag before reaching this threshold compared with WT. Welch's t-test: * $p < 0.05$; *** $p < 0.001$.

Minor comments:

•Line 51: Please revise the phrase “can modulate mating signal the efficacy” to “can modulate the efficacy of mating signals.”

•Line 58: Please correct “bind receptors with highly specificity” to “bind their receptors with high specificity.”

Response:

We have revised the text accordingly. Specifically, the phrasing in line 51 has been corrected to “can modulate the efficacy of mating signals,” and the wording in line 58 has been revised to “bind their receptors with high specificity.” (Page 3, line 23; Page 3, line 31)

Reviewer #2

We thank Reviewer #2 for the careful reading of the manuscript and for the thoughtful comments regarding the conceptual integration of P6 and T2 mutation analyses. We agree that improving the coherence between these two aspects is important, and have revised the manuscript accordingly.

Comment 1:

The only issue I have is that the P6 and T2 stories felt somewhat disconnected. On the bigger scheme, what would be the implications of multiple positions that show such a pleiotropic effect? What could be the mechanistically reason for these positions displaying such effect? While the observations were certainly interesting, it leaves much to be clarified.

Response:

We thank the reviewer for this thoughtful comment and agree that, in the original version, the P6 and T2 results were not sufficiently integrated at the conceptual level. We have substantially revised the manuscript to explicitly frame these findings within a unified functional and evolutionary model of the M-factor peptide.

Specifically, we propose that different positions within the short, structurally constrained M-factor peptide play distinct but complementary roles. P6 represents an environmentally responsive site for which activity is gated by external pH conditions, whereas T2 functions as a permissive or buffering site that expands mutational tolerance and enables otherwise deleterious substitutions to be accommodated. We have revised the Introduction and Results sections to clarify this distinction and the biological implications, and have added a unifying discussion describing how such functionally specialized positions can jointly shape context-dependent fitness trade-offs and evolutionary trajectories.

Mechanistically, we emphasize that these pleiotropic effects likely arise from different modes of influence on pheromone signaling: environment-sensitive receptor activation in the case of P6, and modulation of overall signaling robustness or pathway activation thresholds in the case of T2. Although the precise molecular mechanisms remain to be elucidated, this framework provides a coherent explanation for how multiple positions within a minimal peptide contribute differently to environmental responsiveness, fitness trade-offs, and evolutionary buffering.

We believe that this reorganization clarifies the broader implications of our findings and resolves the perceived disconnection between P6 and T2 analyses. These

changes are reflected in the revised Introduction and Results sections describing the P6 and T2 variants, as well as in the Discussion section throughout the manuscript (Page 3, lines 45–46; Page 4, lines 61–62; Page 8, lines 181–184; Page 9, lines 225–229; Page 12, lines 293–298; Page 13, lines 327–331).

Other minor comments - the authors refer to pH 7 as alkaline multiple times throughout the text. While subsequent pH range experiments did test true alkaline pH ranges, the initial screen goes only up to pH 7, and the juice tests, apple for example, pH 6.5, are not alkaline. The authors should be more precise with these wordings.

Response:

We agree with the reviewer and have revised the manuscript to include more precise pH terminology throughout. References to pH 7.0 and juice-based media (pH ~6.5–7.0) have been changed from “alkaline” to “neutral,” “near-neutral,” or “higher pH,” as appropriate. The term “alkaline” is now used only for conditions at $\text{pH} \geq 7.5$, where true alkaline effects were experimentally tested (see, for example, Page 8, line 188).

Reviewer #3

We thank Reviewer #3 for their detailed and insightful comments, particularly regarding the role of gene redundancy and the genetic background used in our experiments. We agree that addressing this point is essential to strengthen the evolutionary interpretation of our findings.

Comment 1:

*I have one major problem with the logic of the paper: On one hand, it is argued that gene redundancy creates a "safe zone" for variation. Yet, all the experiments are performed with strains that only have one out of the three natural M-factor genes. Hence, it is unclear whether the observed changes in mating behavior are "dominant" or if they would be obscured by two additional wild-type genes. To complicate matters further, the three *mfm* genes are themselves pheromone stimulated to various degrees. It is striking that all three genes encode identical peptides in the wild type. How come that one of these genes have not become this "steppingstone" towards mating performance at a different pH? To address this issue – at least to a certain degree – on of the most significantly different mutants (e.g. P6H) should be introduced into a fully wild-type background harboring the two other (unmutated) *mfm* genes, and then its mating frequency should be compared to a strain with three wild-type genes.*

Response 1:

We thank the reviewer for raising this important conceptual concern regarding gene redundancy and the dominance of pheromone variants. We agree that, in the original version of the manuscript, it was unclear whether the context-dependent effects observed for M-factor variants would persist or be masked in the presence of additional wild-type *mfm* genes.

To address this point, we performed an additional experiment as suggested by the reviewer. Specifically, we introduced either a wild-type *mfm1*⁺ allele or the pH-responsive *mfm1-P36H* allele into an *h⁹⁰* strain retaining intact endogenous *mfm2*⁺ and *mfm3*⁺ genes, thereby generating backgrounds harboring multiple M-factor genes. We then compared mating efficiencies on SSA at pH 5.5 and pH 7.0 with those of a control strain lacking *mfm1* (TS307) and a strain carrying three wild-type *mfm* genes.

Under these conditions, the strain carrying *mfm1-P36H* showed a modestly reduced mating efficiency compared to the strain with three wild-type *mfm* genes at pH 5.5, but was indistinguishable from the strain lacking *mfm1* (TS307) on SSA at both pH 5.5 and pH 7.0 (Extended Data Fig. 8). Thus, the presence of a single P6H allele did not

confer a dominant effect on mating behavior when additional wild-type pheromone genes were present.

These results indicate that redundancy among M-factor genes can buffer or mask the phenotypic effects of environmentally responsive variants, such that their functional consequences become attenuated or cryptic in fully redundant genetic backgrounds. Importantly, this observation does not contradict our conclusion that gene redundancy provides a “safe zone” for variation; rather, it supports a model in which context-dependent variants such as P6H can persist neutrally under redundancy and become phenotypically apparent only when expressed alone or when pheromone signaling becomes limiting. We have incorporated this clarification into the Results, Discussion, and Methods sections and present the new data in Extended Data Fig. 8 (Page 9, lines 214–222; Page 13, lines 348–357; Page 20, lines 562–569; Extended Data Fig. 8).

Extended Data. Fig.8

Extended Data Fig. 8 | Effect of P6H in a wild-type M-factor background.

The mating efficiencies of strains harboring two wild-type M-factor genes (*mfm2⁺* and *mfm3⁺*; TS307) or carrying an additional wild-type *mfm1⁺* (TS1373) or *mfm1-P36H* (TS1374) gene at the *ade6* locus of TS307 were measured on SSA at pH 5.5 and pH 7.0. No significant differences were observed among the strains under either condition, indicating that the P6H variant did not interfere with mating driven by the endogenous wild-type pheromone genes. Welch’s t-test: **p* <0.05; ***p* <0.01; n.s., not significant.

Minor points:

L42: “evolve” -> “can evolve” (*this study suggests but does not prove that this is how it works*).

Response:

We have revised the wording as suggested to “can evolve” to avoid an overstatement (Page 2, line 14).

L50: “*Environmental factors in...*” *There is a problem with the grammar in this sentence.*

Response:

The sentence has been revised to correct the grammatical issue (Page 3, lines 22–24).

L134: *The explanation of amplification or sequencing bias seems unlikely. Also, this could easily be tested experimentally.*

Response:

We agree with the reviewer that attributing the observed frequency changes primarily to amplification or sequencing bias was not sufficiently justified. We have therefore revised the text to adopt a more cautious interpretation. Specifically, we now describe the minor frequency fluctuations observed in control conditions as potentially reflecting subtle differences in vegetative growth or stochastic variation inherent to pooled assays, without invoking technical bias. For the V5M variant, we emphasize that its consistent depletion across conditions is most consistent with a general fitness disadvantage, while noting that technical contributions cannot be completely excluded. The relevant text has been revised accordingly (Page 6, lines 107–113).

L324: *The authors tend to cite their own work rather than the original references. For instance, it was shown already in 1994 (PMID8196631) that the three M-factor genes are identical.*

Response:

We thank the reviewer for pointing this out. We have revised the manuscript to include the original study demonstrating that the three M-factor genes encode identical peptides (PMID: 8196631), and have updated the citations accordingly (Page 13, line 345).

Response to Reviewers

Reviewer #1

We thank the reviewer for the careful evaluation of our manuscript and for the positive assessment of the conceptual framework and experimental design of this study. We appreciate the reviewer's recognition of the evidence supporting environmentally contingent activation of M-factor variants, the antagonistic pleiotropy associated with substitutions at T2, and the permissive role of the T2Q mutation in expanding mutational tolerance.

Comment 1:

Statistical analyses are generally appropriate, with biological replicates clearly defined and standard tests applied. I suggest reporting exact p-values for key comparisons (e.g., WT vs. P6H pairwise competition across pH) and, where possible, emphasizing effect sizes alongside significance. Methodological detail is sufficient to allow reproducibility, and data transparency is good.

Overall, this is a well-executed and conceptually meaningful study that will be of interest to researchers studying signal evolution, cryptic variation, and ligand–receptor diversification.

Response:

In response to the reviewer's suggestion, we have added exact *p*-values for key comparisons and clarified effect sizes where appropriate. Specifically, exact *p*-values are now reported in the figure legends for the WT versus P6H pairwise competition across pH (Fig. 3c,d) and for the relevant comparisons in Fig. 5b. In addition, effect sizes are now explicitly in the Results section (Page 8, lines 196–197), including the change in P6H representation across the pH gradient (Fig. 3d).